# Concordance and discordance of sequence survey methods for molecular epidemiology

Eduardo Castro-Nallar[1], Nur A. Hasan[2,3], Thomas A. Cebula[2,4], Rita R. Colwell[2,3,5], Richard A. Robison[6], W. Evan Johnson[7] and Keith A. Crandall[1]

[1] Computational Biology Institute, George Washington University, Ashburn, VA, USA
[2] CosmosID, College Park, MD, USA
[3] University of Maryland Institute for Advanced Computer Studies, University of Maryland, College Park, MD, USA
[4] Department of Biology, Johns Hopkins University, Baltimore, MD, USA
[5] Bloomberg School of Public Health, Johns Hopkins University, Baltimore, MD, USA
[6] Department of Microbiology and Molecular Biology, Brigham Young University, Provo, UT, USA
[7] Division of Computational Biomedicine, Boston University School of Medicine, Boston, MA, USA

Corresponding author
Eduardo Castro-Nallar, ecastron@gwu.edu

## ABSTRACT

The post-genomic era is characterized by the direct acquisition and analysis of genomic data with many applications, including the enhancement of the understanding of microbial epidemiology and pathology. However, there are a number of molecular approaches to survey pathogen diversity, and the impact of these different approaches on parameter estimation and inference are not entirely clear. We sequenced whole genomes of bacterial pathogens, *Burkholderia pseudomallei*, *Yersinia pestis*, and *Brucella spp.* (60 new genomes), and combined them with 55 genomes from GenBank to address how different molecular survey approaches (whole genomes, SNPs, and MLST) impact downstream inferences on molecular evolutionary parameters, evolutionary relationships, and trait character associations. We selected isolates for sequencing to represent temporal, geographic origin, and host range variability. We found that substitution rate estimates vary widely among approaches, and that SNP and genomic datasets yielded different but strongly supported phylogenies. MLST yielded poorly supported phylogenies, especially in our low diversity dataset, i.e., *Y. pestis*. Trait associations showed that *B. pseudomallei* and *Y. pestis* phylogenies are significantly associated with geography, irrespective of the molecular survey approach used, while *Brucella spp.* phylogeny appears to be strongly associated with geography and host origin. We contrast inferences made among monomorphic (clonal) and non-monomorphic bacteria, and between intra- and inter-specific datasets. We also discuss our results in light of underlying assumptions of different approaches.

## INTRODUCTION

Genomic data coupled with phylogenetic methods have enhanced the ability to track infectious disease epidemics through space and time (*Baker, Hanage & Holt, 2010*). For example, studies have tracked and characterized epidemics occurring at different geographic scales, across local, regional, global, and even historical scales; investigating multidrug-resistant *Staphylococcus aureus* in hospital settings (*Kos et al., 2012*; *Köser et al., 2012*), inferring continental origins of food pathogens (*Goss et al., 2014*), explaining seasonal influenza dynamics (*Lemey et al., 2014*), and ancient oral pathogens (*Warinner et al., 2014*), respectively. Such studies provide valuable information regarding migration rates, directionalities of spread, unique variants, genetic diversity, and drug resistance, as well as informing policy-makers about infection patterns associated with human activities (*Bos et al., 2011*; *Morelli et al., 2010*; *Zhang et al., 2010*). Accordingly, applications of analytical tools to large datasets are abundant in clinical pathology, bioforensics, biosurveillance, and molecular epidemiology (*Reimer et al., 2011*; *Wilson, Allard & Brown, 2013*).

Whole-genome sequencing (WGS) has become an affordable approach for such studies (*Bertelli & Greub, 2013*; *Chen et al., 2013*; *Cornejo et al., 2013*; *Croucher et al., 2013*; *Pérez-Lago et al., 2013*; *Sheppard et al., 2013*; *Wielgoss et al., 2013*). New technologies make it possible to compile datasets that were not even dreamed of twenty years ago (*Chewapreecha et al., 2014*; *Marttinen et al., 2012*; *Nasser et al., 2014*; *Sheppard et al., 2013*) which, in turn, is prompting scientists to ask new questions regarding pathogen distribution, diversity, identification, origin, and phenotype (*Butler et al., 2013*; *Castillo-Ramirez et al., 2012*; *Grad & Waldor, 2013*; *Holt et al., 2012*; *Hong et al., 2014*; *Spoor et al., 2013*).

Because there are now a variety of molecular survey approaches (whole genome sequencing (WGS), multi-locus sequence typing (MLST), and single nucleotide polymorphism (SNP) data) with different costs and resolution abilities, we explored the impact of these different approaches on inferences of population dynamics, transmission patterns, and parameter estimation. For instance, tracking the origin of bioterrorism agents depends on identifying diagnostic mutations, as in the anthrax attacks of 2001 (*Read et al., 2002*), or accurately identifying the subspecies of origin (*Hong et al., 2014*), and understanding the extent to which sampling strategy and choice of molecular survey approach affects temporal and spatial inferences.

Here, we set out to investigate how molecular survey approaches compare, using three select agents as models, namely *Yersinia pestis* (causative agent of plague), *Burkholderia pseudomallei* (causative agent of melioidosis), and *Brucella* spp. (febrile disease). These bacterial species are relevant from health and biosecurity perspectives, and there exists a sizable amount of genomic and supporting information (date of collection, geographic location, and host) for them. Also, they allow for interesting contrasts including comparing intraspecific datasets (*Y. pestis* v. *B. pseudomallei*), one from monomorphic bacteria (clonal), and the other from polymorphic bacteria, as well as interspecific comparisons (*Y. pestis* and *B. pseudomallei* vs. *Brucella* spp.)

Thus, we present and analyze new draft genomic sequences for 20 *Brucella* spp., 20 *Y. pestis*, and 20 *B. pseudomallei* isolates, which we combine with publicly available

genomes (totaling 115 genomes) to compare inferences on evolutionary relationships, dates and rates, and geographic and host structure. Do molecular survey approaches, as currently practiced, produce incongruent inferences? We performed a comparison with real-world examples using species that represent genetic diversities of relevance to clinical molecular epidemiology. We applied different molecular survey approaches (WGS, SNPs, MLST) to evaluate whether these can recover equivalent evolutionary relationships, evolutionary rates and divergence dates, and whether phylogenies inferred with these approaches represent equivalent geographic and host structures.

## METHODS

### Strain selection and sequencing

DNA was isolated from 20 strains of *Burkholderia pseudomallei, Yersinia pestis*, and *Brucella* spp. from the Brigham Young University Select Agent Archive. Samples were selected for sequencing to provide a range of (1) time of isolation, (2) geographic spread, and (3) host association (Table 1). DNA isolation followed standard protocols for select agents and was conducted at the Brigham Young University BSL-3 facility. All DNA preparations received a Certification of Sterility (10% of the final DNA preparation from each isolate was plated for sterility on appropriate agar, and after a minimum of five days of incubation at 37 °C, the samples showed no growth, indicating they contained no viable organisms) before being prepared for sequencing.

The DNA samples were prepared for multiplexed (single-end, 82 cycles) sequencing using a Illumina GAIIx genome analyzer (Illumina Inc., San Diego, CA). For each isolate, genomic library preparations were generated using a Nextera DNA Sample Prep Kit. Post-library quality control and quantification was done using BioAnalyzer 2100 high-sensitivity chips and KAPA SYBR FAST Universal 2X qPCR Master Mix. Post processing of reads was performed by the RTA/SCS v1.9.35.0 and CASAVA 1.8.0. Reads were trimmed to the Q30 level using CLCBio's quality_trim program, and CutAdapt v0.95 was used to excise adapter and transposon contamination.

All sequencing run data and metadata were deposited in the Sequence Read Archive (SRA) under three projects, SRP022877, SRP022862, and SRP023117 for *Y. pestis*, *Brucella* spp., and *B. pseudomallei*, respectively.

### Dataset collection

Short reads were quality filtered (average read quality >30 Phred) and mapped against reference genomes employing the Burrows–Wheeler Transform algorithm, as implemented in SOAP (*Li et al., 2008*). The resulting SAM/BAM files were filtered for duplicate reads that might have arisen by PCR, and consensus sequences were called in Geneious 6.1.6 (*Kearse et al., 2012*; *Li et al., 2009*). Additionally, we retrieved full genomes along with host, collection date, and country of origin metadata for *B. pseudomallei* (11), *Brucella spp.* (18) and *Y. pestis* (26) from GenBank, GOLD, IMG, Patric, Broad Institute, and Pathema databases and resources totaling 115 genomes (Table 1; geographic distribution in Map S1)

**Table 1** **Summary of genomes sequenced and collected in this study.** Metadata on strain source, host, location and date of collection also provided when available.

| NCBI Accession number | Species | Strain | Source | Host | Location | Date of collection |
|---|---|---|---|---|---|---|
| SRX286342 | *Burkholderia pseudomallei* | 5 | Public Health Laboratory Service, London | Sheep | Australia | 1949 |
| SRX286347 | *Burkholderia pseudomallei* | 6 | Public Health Laboratory Service, London | Human | Bangladesh | 1960 |
| SRX286346 | *Burkholderia pseudomallei* | 9 | Public Health Laboratory Service, London | Human | Pakistan | 1988 |
| SRX286345 | *Burkholderia pseudomallei* | 18 | Public Health Laboratory Service, London | Monkey | Indonesia | 1990 |
| SRX286357 | *Burkholderia pseudomallei* | 24 | Public Health Laboratory Service, London | Horse | France | 1976 |
| SRX286354 | *Burkholderia pseudomallei* | 25 | Public Health Laboratory Service, London | Soil | Madagascar | 1977 |
| SRX286353 | *Burkholderia pseudomallei* | 31 | Public Health Laboratory Service, London | Water drain | Kenya | 1992 |
| SRX286352 | *Burkholderia pseudomallei* | 33 | Public Health Laboratory Service, London | Manure | France | 1976 |
| SRX286350 | *Burkholderia pseudomallei* | 35 | Public Health Laboratory Service, London | Human | Vietnam | 1963 |
| SRX286348 | *Burkholderia pseudomallei* | 68 | Public Health Laboratory Service, London | Human | Fiji | 1992 |
| SRX286359 | *Burkholderia pseudomallei* | 91 | Public Health Laboratory Service, London | Sheep | Australia | 1984 |
| SRX286361 | *Burkholderia pseudomallei* | 104 | Public Health Laboratory Service, London | Goat | Australia | 1990 |
| SRX286363 | *Burkholderia pseudomallei* | 208 | Public Health Laboratory Service, London | Human | Ecuador | 1990 |
| SRX286364 | *Burkholderia pseudomallei* | 4075 | Public Health Laboratory Service, London | Human | Holland | 1999 |
| SRX286418 | *Burkholderia pseudomallei* | Darwin-035 | Royal Darwin Hospital | Human | Australia | 2003 |
| SRX286420 | *Burkholderia pseudomallei* | Darwin-051 | Royal Darwin Hospital | Dog | Australia | 1992 |
| SRX286421 | *Burkholderia pseudomallei* | Darwin-060 | Royal Darwin Hospital | Pig | Australia | 1992 |
| SRX286422 | *Burkholderia pseudomallei* | Darwin-077 | Royal Darwin Hospital | Bird | Australia | 1994 |
| SRX286423 | *Burkholderia pseudomallei* | Darwin-150 | Royal Darwin Hospital | Soil | Australia | 2006 |
| SRX286344 | *Burkholderia pseudomallei* | 80800117 | Utah Department of Health | Human | USA | 2008 |
| NC_017832.1 NC_017831.1 | *Burkholderia pseudomallei* | 1026b | hhayden@u.washington.edu | Human | Thailand | 1993 |
| NC_009078.1 NC_009076.1 | *Burkholderia pseudomallei* | 1106a | J. Craig Venter Institute (JCVI) | Human | Thailand | 1993 |

Table 1 (*continued*)

| NCBI Accession number | Species | Strain | Source | Host | Location | Date of collection |
|---|---|---|---|---|---|---|
| NC_012695.1 | *Burkholderia pseudomallei* | MSHR346 | Joint Genome Institute/LANL Center | Human | Australia | 1996 |
| NC_006351.1 NC_006350.1 | *Burkholderia pseudomallei* | k96243 | Sanger Institute | Human | Thailand | 1996 |
| NC_018529.1 NC_018527.1 | *Burkholderia pseudomallei* | BPC006 | Third Military Medical University | Human | China | 2008 |
| NZ_CM000774.1 NZ_CM000775.1 | *Burkholderia pseudomallei* | 1106b | J. Craig Venter Institute (JCVI) | Human | Thailand | 1996 |
| NZ_CM000833.1 NZ_CM000832.1 | *Burkholderia pseudomallei* | 1710a | J. Craig Venter Institute (JCVI) | Human | Thailand | 1996 |
| NC_007435.1 NC_007434.1 | *Burkholderia pseudomallei* | 1710b | J. Craig Venter Institute (JCVI) | Human | Thailand | 1999 |
| NC_009074.1 NC_009075.1 | *Burkholderia pseudomallei* | 668 | J. Craig Venter Institute (JCVI) | Human | Australia | 1995 |
| NZ_CM001156.1 NZ_CM001157.1 | *Burkholderia pseudomallei* | Bp22 | Genome Institute of Singapore | Human | Singapore | 1989 |
| NC_007651 NC_007650 | *Burkholderia thailandensis* | E264 | J. Craig Venter Institute (JCVI) | Soil | Thailand | 1994 |
| SRX278648 | *Brucella abortus* | 1004, Strain 2032 | National Animal Disease Center | Bovine | Missouri, USA | 1990 |
| SRX278790 | *Brucella abortus* | 1007, Strain 2045 | National Animal Disease Center | Bovine | Florida, USA | 1990 |
| SRX278791 | *Brucella abortus* | 1019, Strain 2038 | National Animal Disease Center | Bovine | Tennessee, USA | 1990 |
| SRX278792 | *Brucella abortus* | 1022, Strain 2073 | National Animal Disease Center | Bovine | Georgia, USA | 1990 |
| SRX278793 | *Brucella abortus* | 1146, Strain 8-953 | National Animal Disease Center | Elk | Montana, USA | 1992 |
| SRX278794 | *Brucella abortus* | 1668, Strain 00-666 | National Animal Disease Center | Elk | Wyoming, USA | 2000 |
| SRX278891 | *Brucella abortus* | YELL-99-067 | Idaho National Engineering and Environmental Laboratory | Bison (amniotic fluid) | Wyoming, USA | 1999 |
| SRX282032 | *Brucella abortus* | 1614, Strain Weinheimer 4 | National Animal Disease Center | Bovine | Texas, USA | 2000 |
| SRX282039 | *Brucella canis* | 1107, Strain 1-107 | National Animal Disease Center | Canine | Missouri, USA | 1990 |
| SRX282040 | *Brucella melitensis* | 1253, Strain Ether, L657 | National Animal Disease Center | Caprine | Unknown | 1994 |
| SRX282041 | *Brucella melitensis* | BA 4837 | New Mexico Department of Health | Human | New Mexico, USA | 2003 |
| SRX282042 | *Brucella melitensis* | 70000565 | Utah Department of Health | Blood, Human | Utah, USA | 2000 |
| SRX282044 | *Brucella melitensis* | 80600020 | Utah Department of Health | Blood, Human | Utah, USA | 2006 |
| SRX282045 | *Brucella melitensis* | 80800076 | Utah Department of Health | Human | California, USA | 2008 |
| SRX282046 | *Brucella neotomae* | 1156, Strain 5K33, ATCC#23459 | National Animal Disease Center | Desert wood rat | Unknown, USA | 1992 |
| SRX282047 | *Brucella ovis* | 1117, Strain 1-507 | National Animal Disease Center | Ovine | Georgia, USA | 1991 |

Table 1 (*continued*)

| NCBI Accession number | Species | Strain | Source | Host | Location | Date of collection |
|---|---|---|---|---|---|---|
| SRX282048 | *Brucella ovis* | 1698, Strain 13551-2114; 1985:Dhyatt | National Animal Disease Center | Ovine (semen) | Ft. Collins, Colorado, USA | 2001 |
| SRX282050 | *Brucella species* | 70100304 | Utah Department of Health | Blood, Human | USA- Utah | 2001 |
| SRX282053 | *Brucella suis* | 1103, Strain 2483 | National Animal Disease Center | Porcine | South Carolina, USA | 1990 |
| SRX282057 | *Brucella suis* | 1108, Strain 1-138 | National Animal Disease Center | Porcine | New Jersey, USA | 1990 |
| NC_016777.1 NC_016795.1 | *Brucella abortus* | A13334 | Macrogen | Bovine | Korea | Unknown |
| NC_006932.1 NC_006933.1 | *Brucella abortus* | bv 1, 9-941 | USDA | Bovine | Wyoming, USA | Unknown |
| NC_010740.1 NC_010742.1 | *Brucella abortus* | S19 | Crasta OR | Bovine | Unknown, USA | 1923 |
| NC_010103.1 NC_010104.1 | *Brucella canis* | ATCC 23365 | DOE Joint Genome Institute | Dog | Unknown | Unknown |
| NC_016796.1 NC_016778.1 | *Brucella canis* | HSK A52141 | National Veterinary Research and Quarantine | Dog | South Korea | Unknown |
| NC_012442.1 NC_012441.1 | *Brucella melitensis* | ATCC 23457 | Los Alamos National Lab | Human | India | 1963 |
| NC_017244.1 NC_017245.1 | *Brucella melitensis* | M28 | Chinese National Human Genome Center at Shanghai | Sheep | China | 1955 |
| NC_003317.1 NC_003318.1 | *Brucella melitensis* | bv 1, 16M | Integrated Genomics Inc | Goat | Unknown, USA | Unknown |
| NC_007618.1 NC_007624.1 | *Brucella melitensis* | bv. 1 Abortus 2308 | Lawrence Livermore National Lab | Standard laboratory strain | Unknown | Unknown |
| NC_017246.1 NC_017247.1 | *Brucella melitensis* | M5-90 | Chinese National Human Genome Center at Shanghai | Standard laboratory strain | Unknown | Unknown |
| NC_017248.1 NC_017283.1 | *Brucella melitensis* | bv. 3 NI | China Agricultural University | Bovine | Inner Mongolia, China | 2007 |
| CP001578.1 CP001579.1 | *Brucella microti* | CCM 4915 | Sudic S | Vole | Czech Republic | 2000 |
| NC_009505.1 NC_009504.1 | *Brucella ovis* | ATCC 25840 | J. Craig Venter Institute | Sheep | Australia | 1960 |
| NC_015858.1 NC_015857.1 | *Brucella pinnipedialis* | B2/94 | Zygmunt, M.S. | Seal | UK | 1994 |
| NC_016775.1 NC_016797.1 | *Brucella suis* | VBI22 | Harold R. Garner | Bovine, milk | Texas, USA | Unknown |
| NC_004311.2 NC_004310.3 | *Brucella suis* | bv 1, 1330 | J. Craig Venter Institute | Pig | Unknown, USA | 1950 |

Table 1 (*continued*)

| NCBI Accession number | Species | Strain | Source | Host | Location | Date of collection |
|---|---|---|---|---|---|---|
| NC_010167.1 NC_010169.1 | *Brucella suis* | ATCC 23445 | Joint Genome Institute/LANL Center | Hare | UK | 1951 |
| NC_009667.1 NC_009668.1 | *Ochrobactrum anthropi* | ATCC 49188 | DOE Joint Genome Institute | Arsenical cattle-dipping fluid | Australia | 1988 |
| SRX282065 | *Yersinia pestis* | 4954 | New Mexico Department of Health | Human | NM, USA | 1987 |
| SRX282089 | *Yersinia pestis* | 1901b | New Mexico Department of Health | Human | NM, USA | 1983 |
| SRX282090 | *Yersinia pestis* | Java (D88) | Michigan State University | Unknown | Far East | Unknown |
| SRX282091 | *Yersinia pestis* | Kimberley (D17) | Michigan State University | Unknown | Near East | Unknown |
| SRX282092 | *Yersinia pestis* | KUMA (D11) | Michigan State University | Unknown | Manchuria, China | Unknown |
| SRX282093 | *Yersinia pestis* | TS (D5) | Michigan State University | Unknown | Far East | Unknown |
| SRX282094 | *Yersinia pestis* | 8607116 | New Mexico Department of Health | Dog | NM, USA | Unknown |
| SRX282095 | *Yersinia pestis* | 1866 | New Mexico Department of Health | Squirrel | NM, USA | Unknown |
| SRX282096 | *Yersina pestis* | 4139 | New Mexico Department of Health | Cat | NM, USA | 1995 |
| SRX286281 | *Yersinia pestis* | 4412 | New Mexico Department of Health | Human | NM, USA | 1991 |
| SRX286283 | *Yersinia pestis* | 2965 | New Mexico Department of Health | Human | NM, USA | 1995 |
| SRX286290 | *Yersinia pestis* | 2055 | New Mexico Department of Health | Human | NM, USA | 1998 |
| SRX286302 | *Yersinia pestis* | 2106 | New Mexico Department of Health | Human | NM, USA | 2001 |
| SRX286303 | *Yersinia pestis* | 2772 | New Mexico Department of Health | Cat | NM, USA | 1984 |
| SRX286304 | *Yersinia pestis* | 3357 | New Mexico Department of Health | Mountain lion | NM, USA | 1999 |
| SRX286305 | *Yersinia pestis* | AS 2509 | New Mexico Department of Health | Rodent | NM, USA | 2004 |
| SRX286306 | *Yersinia pestis* | AS 200900596 | New Mexico Department of Health | Rabbit, liver/spleen | United States, Santa Fe, NM | 2009 |
| SRX286307 | *Yersinia pestis* | V-6486 | New Mexico Department of Health | Llama | Las Vegas, NM, USA | Unknown |
| SRX286340 | *Yersinia pestis* | KIM (D27) | Michigan State University | Human | Iran/Kurdistan | 1968 |
| SRX286341 | *Yersinia pestis* | AS200901509 | New Mexico Department of Health | Liver/spleen, prairie dog | Santa Fe, NM, USA | 2009 |
| NC_017168.1 | *Yersinia pestis* | A1122 | Los Alamos National Lab | Ground squirrel | California | 1939 |
Table 1 (*continued*)

| NCBI Accession number | Species | Strain | Source | Host | Location | Date of collection |
|---|---|---|---|---|---|---|
| NC_010159.1 | *Yersinia pestis* | Angola | J. Craig Venter Institute (JCVI) | Human | Angola | Unknown |
| NC_008150.1 | *Yersinia pestis* | Antiqua | DOE Joint Genome Institute | Human | Congo | 1965 |
| PRJNA54473 | *Yersinia pestis* | B42003004 | J. Craig Venter Institute (JCVI) | Marmota baibacina | China | 2003 |
| PRJNA54563 | *Yersinia pestis* | CA88-4125 | DOE Joint Genome Institute | Human | California | 1988 |
| NC_003143.1 | *Yersinia pestis* | CO92 | Sanger Institute | Human/cat | Colorado | 1992 |
| NC_017154.1 | *Yersinia pestis* | D106004 | Chinese Center for Disease Control and Prevention | Apodemus chevrieri | Yulong County, China | 2006 |
| NC_017160.1 | *Yersinia pestis* | D182038 | Chinese Center for Disease Control and Prevention | Apodemus chevrieri | Yunnan, China | 1982 |
| PRJNA54471 | *Yersinia pestis* | E1979001 | J. Craig Venter Institute (JCVI) | Eothenomys miletus | China | 1979 |
| PRJNA54469 | *Yersinia pestis* | F1991016 | J. Craig Venter Institute (JCVI) | Flavus rattivecus | China | 1991 |
| PRJNA54399 | *Yersinia pestis* | FV-1 | The Translational Genomics Research Institute | Prairy dog | Arizona | 2001 |
| PRJNA55339 | *Yersinia pestis* | India 195 | DOE Joint Genome Institute | Human | India | Unknown |
| PRJNA54383 | *Yersinia pestis* | IP275 | The Institute for Genomic Research | Human | Madagascar | 1995 |
| NC_009708.1 | *Yersinia pseudotuberculosis* | IP31758 | J. Craig Venter Institute (JCVI) | Human | Russia | 1966 |
| PRJNA54475 | *Yersinia pestis* | K1973002 | J. Craig Venter Institute (JCVI) | Marmota himalaya | China | 1973 |
| PRJNA42495 | *Yersinia pestis* | KIM D27 | J. Craig Venter Institute (JCVI) | Human | Iran/Kurdistan | 1968 |
| NC_004088.1 | *Yersinia pestis* | KIM10+ | Genome Center of Wisconsin | Human | Iran/Kurdistan | 1968 |
| NC_017265.1 | *Yersinia pestis* | Medievalis str. Harbin 35 | Virginia Bioinformatics Institute | Human | China | 1940 |
| PRJNA54477 | *Yersinia pestis* | MG05-1020 | J. Craig Venter Institute (JCVI) | Human | Madagascar | 2005 |
| NC_005810.1 | *Yersinia pestis* | Microtus 91001 | Academy of Military Medical Sciences, The Institute of Microbiology and Epidemiology, China | Microtus brandti | China | 1970 |
| NC_008149.1 | *Yersinia pestis* | Nepal516 | Genome Center of Wisconsin | Human/soil | Nepal | 1967 |
| PRJNA55343 | *Yersinia pestis* | Pestoides A | DOE Joint Genome Institute | Human | FSU | 1960 |
| PRJNA58619 | *Yersinia pestis* | Pestoides F | DOE Joint Genome Institute | Human | FSU | Unknown |
| PRJNA55341 | *Yersinia pestis* | PEXU2 | Enteropathogen Resource Integration Center (ERIC) BRC | Rodent | Brazil | 1966 |
| PRJNA54479 | *Yersinia pestis* | UG05-045 | J. Craig Venter Institute (JCVI) | Human | Uganda | 2005 |
| PRJNA47317 | *Yersinia pestis* | Z176003 | CCDC | Marmota himalayana | Tibet | 1976 |

**Table 2 Genetic diversity and dataset length for different species and molecular survey approaches.**

| | | MLST | | SNP | | | | Genome | | | |
| | | | | Chromosome I | | Chromosome II | | Chromosome I | | Chromosome II | |
| | | Mean | Variance | Mean | Variance | Mean | Variance | Mean | Variance | Mean | Variance |
|---|---|---|---|---|---|---|---|---|---|---|---|
| **Bp** | theta | 36.05 | 127.39 | 7807.03 | 5593613.54 | 4333.68 | 1724029.33 | 4666.60 | 1999005.02 | 2187.99 | 439709.21 |
| | pi | 4.45E–03 | 5.15E–06 | 9.58E–02 | 2.18E–03 | 9.69E–02 | 2.24E–03 | 5.67E–03 | 7.66E–06 | 7.14E–03 | 1.21E–05 |
| | length | 3518.00 | | 31189.00 | | 17313.00 | | 289172.00 | | 108654.00 | |
| **Br** | theta | 112.58 | 1205.64 | 914.77 | 78109.73 | 234.25 | 5161.72 | 635.85 | 37782.92 | 2046.37 | 390305.58 |
| | pi | 1.00E–02 | 2.46E–05 | 8.01E–02 | 1.54E–03 | 7.66E–02 | 1.43E–03 | 7.96E–03 | 1.52E–05 | 1.62E–02 | 6.25E–05 |
| | length | 4409.00 | | 3628.00 | | 929.00 | | 24110.00 | | 36223.00 | |
| **Yp** | theta | 36.17 | 120.42 | 3204.65 | 883301.99 | | | 527.40 | 24020.48 | | |
| | pi | 4.97E–04 | 6.68E–08 | 5.60E–02 | 7.40E–04 | | | 4.55E–04 | 4.94E–08 | | |
| | length | 20498.00 | | 14116.00 | | | | 281149.00 | | | |

**Notes.**

Bp, *Burkholderia pseudomallei*; Br, *Brucella spp.*; Yp, *Yersinia pestis.*

(*Benson et al., 2010*; *Brinkac et al., 2010*; *Gillespie et al., 2011*; *Liolios et al., 2008*; *Markowitz et al., 2012*). From the assembled genomes we derived all datasets as described below.

Multi-locus sequence type markers for *B. pseudomallei*, namely ace, *gltB*, *gmhD*, *lepA*, *lipA*, *narK*, and *ndh* were retrieved from the PubMLST database (http://bpseudomallei. mlst.net). For *Brucella* spp., we resorted to markers used by *Whatmore, Perrett & MacMillan (2007)* i.e., *gap*, *aroA*, *glk*, *dnaK*, *gyrB*, *trpE*, *cobQ*, *omp25*, and *int-hyp*. Likewise, for *Y. pestis* we obtained markers from PubMLST (Yersinia spp.; http://pubmlst.org/ yersinia/) *aarF*, *dfp*, *galR*, *glnS*, *hemA*, *rfaE*, and *speA*. In addition, we obtained markers from *Achtman et al. (1999)* (*dmsA*, *glnA*, *manB*, *thrA*, *tmk*, and *trpE*) and from *Revazishvili et al. (2008)* (*16S rDNA*, *gyrB*, *yhsp*, *psaA* and *recA*). We created a custom BLAST (*Altschul et al., 1990*) database with our new genome sequences combined with the publicly available genomes for all three species groups.

We created datasets based on SNPs by searching for $k\text{-}mer = 25$ (SNP on position 13) among unaligned genomes, as implemented in kSNP 2.0 (*Gardner & Hall, 2013*; *Gardner & Slezak, 2010*). We chose this implementation because it does not depend on arbitrarily selecting a reference genome, it can take draft and unassembled sequence data (including low-coverage genomes), and it is fast and widely used in epidemiological studies (*Epson et al., 2014*; *Pettengill et al., 2014*; *Raphael et al., 2014*; *Timme et al., 2013*). Briefly, the optimal *k-mer* size was estimated using Kchooser, which identifies threshold value of *k* for which non-unique *k-mers* are the result of real genome redundancy, not chance. We kept all SNPs that were shared among all taxa in a given dataset (core SNP subset), which were used to build matrices for downstream analyses. The matrices used contained only variable bi-allelic sites from non-overlapping *k-mers* and their size is described in Table 2.

We created full genome datasets by aligning complete genome sequences in Mauve 2.3.1 (*Darling, Mau & Perna, 2010*) and then used the resulting multiple sequence alignment directly and/or reduced for phylogenetic inference. The reduced full genome dataset consisted of all Locally Collinear Blocks (LCBs) detected by Mauve that were greater than

10 Kb and randomly subsampled up to a total of 300 Kb present across all taxa in a given dataset.

### Diversity and phylogenetic analyses

We measured genetic diversity as the substitution rate-scaled effective population size $\Theta$ for all molecular survey approaches (MLST, SNP, WGS), as implemented in the 'pegas' package in R (*Paradis, 2010*). We inferred phylogenies, both with and without assuming a molecular clock. Clock phylogenies were inferred using Bayesian Inference (BI) and Markov Chain Monte Carlo (MCMC) simulations as implemented in Beast 1.7.5 (restricting the analysis to those sequences with recorded dates) and using the Beagle library to speed up analysis (*Ayres et al., 2012*; *Drummond et al., 2012*). We assumed a General Time Reversible (GTR) substitution model for all three data approaches with a discrete gamma distribution (4 categories) to model rate heterogeneity (MLST datasets were partitioned by gene with a model fit per gene; rate heterogeneity was not modeled for SNP datasets). We unsuccessfully tried to partition the genome dataset by gene, but phylogenetic inference did not reach convergence. Briefly, MCMC simulations were run until a single chain reached convergence, as diagnosed by its trace and ESS values ($>400$; ranging from $2E^8$ to $2E^9$ steps; 10% burnin) in Tracer 1.5 (http://tree.bio.ed.ac.uk/software/tracer/) and tree distributions were summarized in TreeAnnotator 1.7.5 (10–20% of trees were discarded as burnin). The molecular clock (strict clock model) was calibrated using isolate collection dates and a uniform distribution (from 0 to 1) as clock prior. We also used BI for non-clock phylogenies as implemented in MrBayes 3.2 (*Ronquist et al., 2012*) where we ran 8 chains (6 heated), $2E^7$ generations each. As in the clock phylogenies, we used visual inspection of the traces as well as the average standard deviation of split frequencies to assess convergence. All trees were rooted by using outgroups (*Yersinia pseudotuberculosis* IP31758, *Ochrobactrum anthropic*, and *Burkholderia thailandensis* E264).

In order to compare tree topologies, we applied two topology metrics, Robinson–Foulds (RF, *Robinson & Foulds, 1981*) and Matching Splits Clusters (MC, *Bogdanowicz & Giaro, 2012*) to compare topologies across different molecular survey approaches and among chromosomes as implemented in TreeCmp (*Bogdanowicz & Giaro, 2012*). We also assessed the extent to which phylogenies and traits (host range, sample collection site, and sampling date) were correlated through Bayesian Tip-Significance testing by estimating the Association Index (AI, *Wang et al., 2001*) and Parsimony Score (PS, *Slatkin & Maddison, 1989*) as implemented in BaTs (*Parker, Rambaut & Pybus, 2008*). Figures were plotted using ggplot2 (*Wickham, 2009*) and APE (*Paradis, Claude & Strimmer, 2004*) packages, and high posterior density (HPD) intervals were estimated using TeachingDemos package (*Snow & Snow, 2013*).

## RESULTS AND DISCUSSION

Sequencing technologies and statistical phylogenetic methods are arming researchers with powerful tools to track infectious agents over space and time with unprecedented

resolution (*Holt et al., 2012*; *Lewis et al., 2010*). However, with multiple molecular survey approaches and a battery of analytical methods, it is not clear how these interact.

Using 115 genomic sequences (60 this study + 55 GenBank), we compared inferences regarding genetic diversity, substitution rates and node ages, tree topologies, structure and phylogenies inferred from different molecular survey approaches. We use the term "molecular survey approaches" to refer to either MLST, SNP, or WGS approaches, and the term "species datasets" or simply "dataset" to refer either to *B. pseudomallei*, *Brucella spp.* or *Y. pestis* sequence data belonging to any of these species/genera. Given the difficulty of current algorithm implementations in reading and analyzing whole bacterial genomes, we decided to randomly sub-sample core homologous regions to compile genomic data that we termed "genome" (see Methods for details).

## Diversity and datasets

Datasets sizes varied in length by data approach, species, and genomic partition (chromosome I/II). Notably, we intended to include as many genes as possible for the MLST schemes, which resulted in partitioned datasets ranging from 7 to 18 genes. In the case of *Y. pestis*, the MLST dataset constituted a larger dataset than the SNP dataset due to the low variability in this species. The interspecific dataset, i.e., *Brucella spp.*, rendered the smallest dataset for all data approaches (least number of sites) as opposed to intraspecific datasets (*Y. pestis*; *B. pseudomallei*) that ended up being one or two orders of magnitude longer (Table 2; square brackets).

In order to characterize the present genetic diversity of our datasets, we estimated effective population size using a segregating sites model ($\Theta$; Watterson's theta) and nucleotide diversity ($\pi$) (*Nei, 1987*; *Paradis, 2010*; *Watterson, 1975*). Nucleotide diversity ranked higher for SNPs compared to other approaches for the same species, as these data only contain binary variable sites (Table 2). Nucleotide diversity was higher for *B. pseudomallei* than for *Brucella spp.* and *Y. pestis* when SNP data were analyzed. However, this was not observed for either MLST or genome data, where nucleotide diversity ranked higher for *Brucella spp.* compared to *B. pseudomallei*. *Y. pestis* nucleotide diversity was consistently lower compared to other datasets across molecular approaches. $\Theta$ estimates were higher for *B. pseudomallei* than others for SNP and genome data, but not MLST data where *Brucella spp.* yielded the larger $\Theta$ (Table 2).

## Rates and ages

We tested whether different data approaches resulted in different inferences regarding substitution rates and node ages while maintaining other parameters constant, i.e., clock calibrations, substitution models, tip dates, coalescent tree priors, and taxa (different partition scheme; see Methods for details). Substitution rate estimates were always higher for SNP data compared to genome data, irrespective of species datasets used (Fig. 1). Rates estimated from MLST data were largely overlapping with estimates from genome data for *Y. pestis* and *Brucella spp.* including median values (highlighted in Figs. 1B–1C). However, this was not the case for the *B. pseudomallei* dataset where, although the distributions overlapped, median values for the substitution rate estimate from MLST data were higher

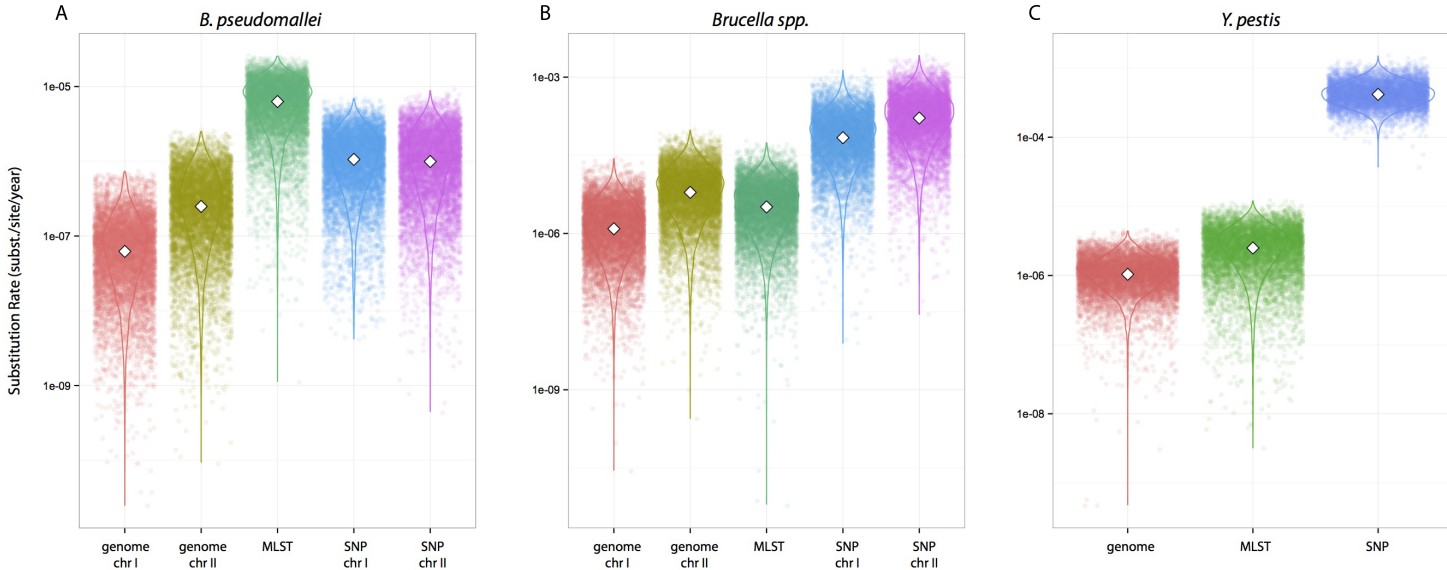

**Figure 1 Substitution rates for all datasets as estimated from different molecular survey approaches.** Genome/SNP chr I/II refers to estimates from different chromosomes. *Burkholderia pseudomallei* (A), *Brucella spp.* (B), and *Yersinia pestis* (C). Note different scale for species rates.

by at least an order of magnitude compared to estimates from other approaches (MLST rate median $= 6.30E^{-7}$; genome chr I $= 6.17E^{-8}$; genome chr II $= 2.48E^{-7}$; SNP chr I $= 1.06E^{-6}$; SNP chr II $= 9.94E^{-7}$ [rates in substitutions per site per year]).

Remarkably, when collecting node ages and comparing them across data approaches, we found that highest posterior density intervals (95% HPD) overlapped substantially in the case of *Y. pestis* and *Brucella spp.* datasets (Figs. 2B–2C). We observed the same trend with SNP and genome approaches when analyzing *B. pseudomallei* datasets, but not with MLST data (Fig. 2A). Interestingly, in *Y. pestis* node age estimates we observed that 95% HPD intervals were narrower in SNP and genome data than in MLST data. This suggests that, though different molecular survey approaches result in markedly different substitution rate estimates, node ages 95% HPD are largely overlapping and thus not significantly different.

Substitution rate estimates differ substantially (up to 2 orders of magnitude), though their posterior distributions overlap to various degrees. Generally, substitution rate estimates drawn from SNP data were higher than those from MLST and genome data. However, node ages are largely consistent across molecular survey approaches, especially for *Brucella spp.* data (interspecific and intermediate diversity dataset). This supports the practice of using SNP data coupled to Bayesian inference coalescent methods to infer divergence times, even though traditional reversible substitution models are not specifically designed for this molecular approach. Substitution models based on models for discrete morphological character changes have been suggested, but are not widely popular (*Lewis, 2001*).

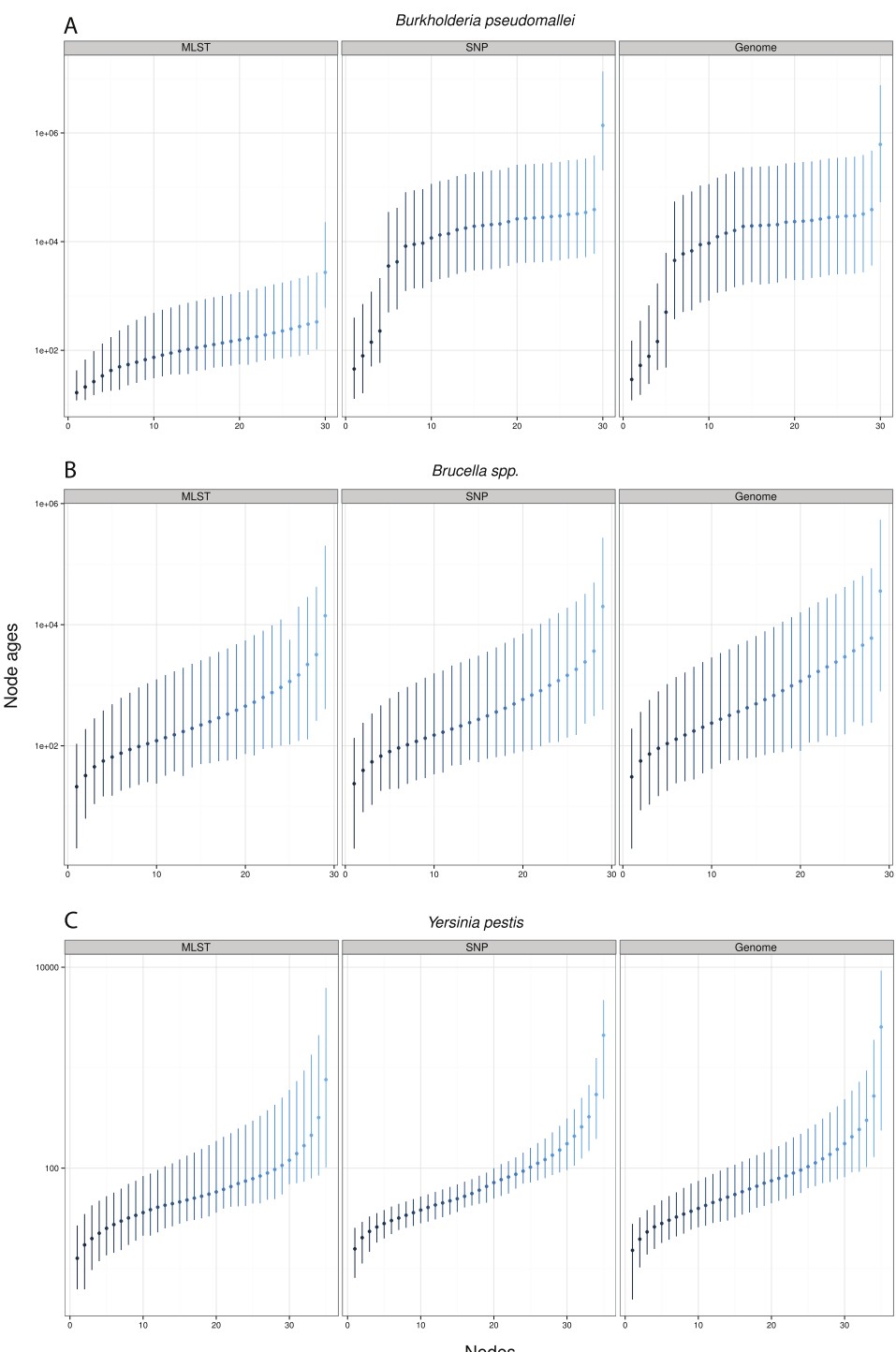

**Figure 2 Median node ages in years.** *Burkholderia pseudomallei* (A), *Brucella spp.* (B), and *Yersinia pestis* (C) median estimates and their 95% highest posterior density (HPD) interval according to molecular survey approach (only chromosome I showed; see Fig. S1). Nodes are numbered from youngest to oldest.

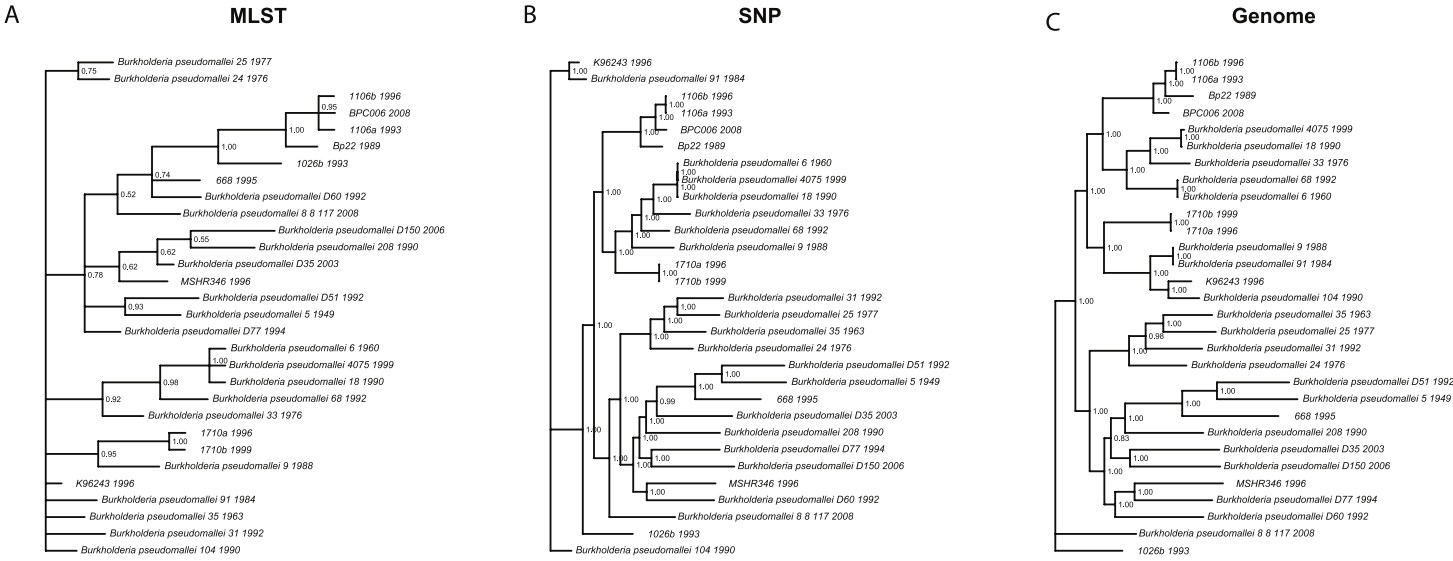

**Figure 3** *Burkholderia pseudomallei* **phylogenies by survey approach.** MLST phylogeny (A) is less resolved and poorly supported compared to SNP (B) and genome (C) phylogenies (only chromosome I showed).

## Phylogenies and topology comparisons

We wanted to determine whether different data approaches produce different phylogenies and to quantify the extent of any observed differences in topology (dataset sizes in Table 2). We inferred phylogenies for every species under all three molecular survey approaches, partitioned by chromosome when appropriate, without assuming a molecular clock and outgroup rooted (see 'Methods'). We used two topology metrics, Matching Clusters (MC), rooted version of Matching splits (*Bogdanowicz & Giaro, 2012*), and R–F Clusters (RC), rooted version of Robinson–Foulds metric (*Bogdanowicz, Giaro & Wróbel, 2012*; *Robinson & Foulds, 1981*). MC distances reflect the minimal number of cluster (or clade) movements needed so that the two phylogenies are topologically equivalent. RC distances measure the average number of cluster differences between two phylogenies. Likewise, MC distances can be interpreted as reflecting changes deep in a phylogeny, and RC distances, in turn, as reflecting changes at the tip of the phylogenies or for more recent relationships. In general, we found that the phylogenies inferred using MLST data are less resolved and more poorly supported (posterior probabilities) than those inferred by either SNP or genome data for all species datasets (Fig. 3 and Fig. S3). This is also reflected in MC distances, where topologies inferred by MLST data are as distant, or more so, to SNP/genome based topologies than between SNP and genome topologies, with some exceptions (Table 3).

For *Brucella spp.* and *Y. pestis*, MC distances are clearly higher between SNP/genome and MLST; however, RC distances do not follow this trend. Since MC metric concentrates more on differences corresponding to branches deep in the topologies as opposed to RC, these results suggest that SNP and genome topologies have more similar backbones when compared to each other than MLST topologies. Likewise, MLST topologies are more similar at the tips rather than deep in the topologies (Fig. 3 and Table 3). Of course, to determine which approach is more accurate would require a dataset of known evolutionary

**Table 3 Topology distances among phylogenies inferred using different molecular survey approaches.** Bolded rows show tree comparisons between different chromosomes under the same molecular survey approach.

| Species | Tree comparisons | | Matching cluster | R–F cluster |
|---|---|---|---|---|
| *B. pseudomallei* | mlst | snp-I | 181 | 16 |
| *B. pseudomallei* | mlst | snp-II | 162 | 17 |
| *B. pseudomallei* | mlst | genome-I | 149 | 18 |
| *B. pseudomallei* | mlst | genome-II | 116 | 18 |
| ***B. pseudomallei*** | **snp-I** | **snp-II** | **33** | **7** |
| *B. pseudomallei* | snp-I | genome-I | 56 | 17 |
| *B. pseudomallei* | snp-I | genome-II | 91 | 17 |
| *B. pseudomallei* | snp-II | genome-I | 47 | 19 |
| *B. pseudomallei* | snp-II | genome-II | 72 | 16 |
| ***B. pseudomallei*** | **genome-I** | **genome-II** | **63** | **14** |
| *Brucella spp.* | mlst | snp-I | 34 | 5 |
| *Brucella spp.* | mlst | snp-II | 10 | 1.5 |
| *Brucella spp.* | mlst | genome-I | 73 | 4.5 |
| *Brucella spp.* | mlst | genome-II | 56 | 5 |
| ***Brucella spp.*** | **snp-I** | **snp-II** | **24** | **5.5** |
| *Brucella spp.* | snp-I | genome-I | 61 | 5.5 |
| *Brucella spp.* | snp-I | genome-II | 40 | 5 |
| *Brucella spp.* | snp-II | genome-I | 63 | 5 |
| *Brucella spp.* | snp-II | genome-II | 50 | 4.5 |
| ***Brucella spp.*** | **genome-I** | **genome-II** | **67** | **7.5** |
| *Y. pestis* | mlst | snp | 223 | 13.5 |
| *Y. pestis* | mlst | genome | 103 | 8 |
| *Y. pestis* | snp | genome | 124 | 8.5 |

**Notes.**
Genome/SNP-I/II, chromosome I or II; R–F Cluster, Robinson–Foulds for rooted trees metric.

history, but SNP and genome approaches appear to be more consistent with one another, especially for the deeper nodes.

Slowly evolving pathogens can be difficult to track as their populations accrue fewer substitutions, and/or genomic changes may or may not reflect ecological processes, such as host switches or geographic spread (see below for association testing). For instance, phylogenies inferred using MLST data were less resolved and poorly supported compared to their SNP and genome counterparts, even though in some cases (e.g., *Brucella* spp./*Y. pestis*) the MLST dataset size was larger than the SNP size dataset. This argues for the need to acquire genome data, as those data constitute the ultimate source of genealogical information, especially when analyzing monomorphic or clonal species, i.e., *Y. pestis* (*Achtman, 2008*; *Achtman et al., 1999*). We also found that strongly supported phylogenies, e.g., those based on SNP and genome data, can support conflicting hypotheses and thus will be misleading. For instance, *B. pseudomallei* clades, including isolates 1106a, 1106b, Bp22, and BPC006, all show posterior probabilities = 1, yet their relationships differ, hence

**Table 4** **Trait-phylogeny association statistics.** Significant associations (*p* value <0.05) were found between traits (sampling location/host/time) and phylogenies inferred by using different data approaches. Association index (AI); Parsimony Score (PS); genome/SNP-I/II, chromosome I or II.

| Statistic | Trait | | |
|---|---|---|---|
| | **Sampling location** | | |
| | *B. pseudomallei* | *Brucella spp.* | *Y. pestis* |
| **AI** | MLST, genome-I, genome-II, SNP-I, SNP-II | MLST, genome-I, genome-II, SNP-I, SNP-II | MLST, genome, SNP |
| **PS** | MLST, genome-I, genome-II, SNP-II | MLST, SNP-II | MLST, genome, SNP |
| | **Host** | | |
| | *B. pseudomallei* | *Brucella spp.* | *Y. pestis* |
| **AI** | None | MLST, genome-I, genome-II, SNP-I, SNP-II | Genome, SNP |
| **PS** | None | MLST, genome-I, genome-II, SNP-I, SNP-II | None |
| | **Time** | | |
| **AI** | Genome-I, genome-II | None | Genome |
| **PS** | None | None | Genome |

a caveat when analyzing SNP/genome data and drawing conclusions about relationships amongst isolates.

## Phylogenetic associations with geography, time, and host

Phylogenetic inference often is performed to infer ecological processes that leave a genomic imprint. Phylogeny-trait associations are essential to elucidate these processes. Accordingly, we estimated the Association Index (AI), and Parsimony Score (PS) on three traits (sampling location, sampling time, and host), and tested whether different answers were obtained by molecular survey. Results for *B. pseudomallei* showed significant association with sampling location and sampling time, but not with host for most of the datasets (AI and PS; Table 4). Likewise, *Y. pestis* datasets were significantly associated with sampling location and, to some extent, with sampling time and host. Interestingly, *Brucella spp.* showed significant genetic structure to be associated with both sampling location and host, but not sampling time (Table 4).

Irrespective of the molecular survey approach used, phylogenies derived from *B. pseudomallei* showed a significant association with sampling location but not with host, suggesting similar evolutionary forces acting on *B. pseudomallei* in different hosts, or that *B. pseudomallei* isolates are highly endemic to the sites from which they were isolated. Similarly, *Brucella spp.* phylogenies were associated with both sampling location and host, irrespective of the data approach used, which most likely reflects metabolic and geographic constraints on gene flow. Interestingly, for *Y. pestis*, no significant association of host and MLST data was observed, which most likely reflects lack of signal, given the absence of resolution of phylogenies in its posterior distribution.

Molecular survey approaches have different sets of assumptions and properties that must be considered before an analysis is done. Similarly, statistical models that are employed may be suited for certain data approaches and not others. Here, we used popular phylogenetic methods for all molecular approaches to test whether congruent inferences

could be obtained, even though some might violate particular model assumptions. The MLST method targets housekeeping genes that are likely to be maintained across taxonomic levels, hence amenable for evolutionary inferences. Yet, similar to other molecular survey approaches, they are likely to be subjected to selective pressures, which may or may not impact evolutionary inferences because of molecular convergence (*Castoe et al., 2009*; *Edwards, 2009*) and estimation of branch lengths (*Ho et al., 2011*; *Roje, 2014*). Other trade-offs of MLST have been discussed elsewhere, mainly with respect to utility and how they can be refashioned in the post-genomic era (*Maiden et al., 2013*; *Pérez-Losada et al., 2013*). On the other hand, sampling bias can influence phylogenetic analysis (*Lachance & Tishkoff, 2013*). Here, we obtained SNP data without using reference data and included globally sampled genomes and stringent quality controls (high Phred scores, long *k-mers*) to diminish ascertainment and discovery bias (*Gonçalves da Silva et al., 2014*). However, standard nucleotide substitution models, such as GTR, are not designed to account for binary sites-only datasets nor Bayesian Inference methods, which typically factor in invariable sites, influencing branch length estimation and impacting parameter estimates such as substitution rate and divergence time. Nonetheless, they have been used to date the spread of bacteria and other pathogens (*Comas et al., 2013*; *Holt et al., 2010*; *Holt et al., 2012*; *Okoro et al., 2012*; *Pepperell et al., 2013*). We speculate, based on these results, that analysis of SNP data to survey genomic variation is robust and can produce inferences that are not substantially different from WGS data. However, a recent study using simulated data has shown that using SNPs and a single reference introduce systematic biases and errors in phylogenetic inference (*Bertels et al., 2014*).

## CONCLUSIONS

The field of bacterial population genomics is advancing rapidly with larger datasets (more taxa, more sites) increasingly available, including whole-genomes, making greater resolution possible and more powerful exploration of complex issues (*Chewapreecha et al., 2014*; *Nasser et al., 2014*). The results of analyses reported here show that the molecular survey that is used can have a critical impact on substitution rate and phylogenetic inference. However, node dates and trait associations are relatively consistent irrespective of the survey tool used. We found substitution rates vary widely depending on the approach taken, and SNP and genomic datasets yield different, but strongly supported phylogenies. Overall, inferences were more sensitive to molecular survey in the low diversity *Y. pestis* dataset, compared to the *B. pseudomallei* and *Brucella spp.* datasets.

Substitution rate estimates are important because, coupled to sampling dates, they allow tracking infections in space and time, and thus provide an essential epidemiological tool for monitoring and control of infectious diseases. The results presented strongly suggest that future studies should consider discordances between inferences derived from different molecular survey methods, especially with respect to substitution rate estimates.

In practice, other variables also influence what type of survey approach to use, and there are foreseeable cases where it might be practical to choose for MLST over WGS/SNPs, e.g., cost, equipment, ease of use, and necessary expertise. More importantly, coupling

multiple molecular survey approaches could be useful in gaining biological insights (e.g., genome evolution, gene synteny, and content using WGS) and genotyping large numbers of samples using MLSTs.

Importantly, for whole genome analysis, a subset of data is selected to run existing software to estimate population genetic parameters. Clearly, there is a need to expand the range of methods to include whole genome data analysis. However, as bacterial genomics matures, current methods will need to be modified and extended to handle the stream of data now being generated.

## ACKNOWLEDGEMENTS

We would like to thank the BYU Select Agent Archive for providing biological specimens. We thank the GW Colonial One computing cluster for compute time for these analyses.

### Funding

The Department of Homeland Security provided funding from Grant# HSHQDC-10-C-00177. Eduardo Castro-Nallar was funded by "CONICYT + PAI/ CONCURSO NACIONAL APOYO AL RETORNO DE INVESTIGADORES/AS DESDE EL EXTRANJERO, CONVOCATORIA 2014 + FOLIO 82140008". The funders had no role in study design, data collection and analysis, decision to publish, or preparation of the manuscript.

### Grant Disclosures

The following grant information was disclosed by the authors:
Department of Homeland Security: HSHQDC-10-C-00177.
CONICYT + PAI/ CONCURSO NACIONAL APOYO AL RETORNO DE INVESTIGADORES/AS DESDE EL EXTRANJERO, CONVOCATORIA 2014 + FOLIO 82140008.

### Competing Interests

Eduardo Castro-Nallar, W. Evan Johnson, and Keith A. Crandall have a combination of ownership of, and employment in, Aperiomics, Inc. Nur A. Hassan, Thomas A. Cebula, and Rita R. Colwell have a combination of ownership of, and employment in, CosmosID.

### Author Contributions

- Eduardo Castro-Nallar conceived and designed the experiments, performed the experiments, analyzed the data, contributed reagents/materials/analysis tools, wrote the paper, prepared figures and/or tables, reviewed drafts of the paper.
- Nur A. Hasan conceived and designed the experiments, performed the experiments, contributed reagents/materials/analysis tools, reviewed drafts of the paper.
- Thomas A. Cebula, Richard A. Robison and W. Evan Johnson contributed reagents/materials/analysis tools, reviewed drafts of the paper.
- Rita R. Colwell conceived and designed the experiments, contributed reagents/materials/analysis tools, reviewed drafts of the paper.

**PeerJ** ______________________________________________________

- Keith A. Crandall conceived and designed the experiments, analyzed the data, contributed reagents/materials/analysis tools, wrote the paper, reviewed drafts of the paper.

## DNA Deposition

The following information was supplied regarding the deposition of DNA sequences:
  GenBank: SRP022877; SRP022862; SRP023117.

## Data Deposition

The following information was supplied regarding the deposition of related data:
  All supplemental information has been deposited in FigShare at: http://dx.doi.org/10.6084/m9.figshare.1091392.

## Supplemental Information

Supplemental information for this article can be found online at http://dx.doi.org/10.7717/peerj.761#supplemental-information.

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
