# Peer review of "Concordance and discordance of sequence survey methods for molecular epidemiology"

_PeerJ, doi:10.7717/peerj.761_

## Round 0.1 · original submission · Minor Revisions

Dear Dr. Castro-Nallan and coauthors,

These revisions will not require additional outside review, but will need to address the comments raised below by the reviewers.

Reviewer 1 ·

Basic reporting

Castro-Nallan et al. undertake a comprehensive effort to show under which conditions gene sampling vs. SNP vs. WGS provide similar or different results in use cases related to pathogen outbreaks. I don't think anything in the manuscript was a surprise, esp. given they are unable to get convergence at single loci re: phylogenetic analysis. WGS > SNP > MLGT, but the nature of the experiments provides an interesting calibration or assessment of these studies in a large (>100 genome) dataset

Experimental design

The k-mer driven SNP portion of the work was unclear, both in terms of the experimental design and how it exactly fits in the surveillance / epidemiological setting at the core of this work.

As I read the kSNP2 paper relative to this manuscript, it is an alignment-free system that can find bi allelic variants flanked by 12 non-variant sites. The implication though is it specifically hones in on variable regions of the genome, so of course the substitution rate estimation will be higher unless I am missing something. The authors should add in text in the methods explaining how/why a researcher would use alignment-free SNPs in epidemiological settings and how it relates to current work in the field. For example, with ever decreasing sequencing costs and next-gen assemblers likely more than capable of assembling genic regions of bacteria genomes, is the SNP approach targeting lower sequencing depth sampling (and therefore higher throughout re: # of isolates/lane)? Or are these researchers/their community doing more metagenomics level analysis where these "signatures" would be useful (ala metaPhylan, Insignia, etc.) since assembly/coverage in these settings may be challenging?

Finally, having worked on multiple insect population genomics efforts focused on bialleilic SNPs I was slightly surprised the authors did not look at principal components analysis (PCA) which is a powerful and mostly assumptionless test to look at relationships with sampling time/location/etc. and general phylogenetic relationships. Is this a consequence of down sampling to 300kb of sequence for the comparisons with WGS approaches? In our settings we typically filter reference-based SNPs (akin to the WGS approach here) in the interest of computational requirements, but given the relatively smaller genome sizes PCA may do better with Y. pestis than the methods discussed here.

Validity of the findings

The work seems well done given the overall experimental design. The authors should discuss bias, if any, inherent in the SNP discovery phase and how that may elevate substitution rate relative to more "fair" approaches based on shared loci.

Additional comments

I think work such as this is important, specifically for those interested in getting involved in genomics methods. Although PCA-analysis may be out-of-scope, I think more discussion in the intro and methods about SNP vs. MLGT given inexpensive sequencing will make this paper more self contained to readers like me who are knowledgeable about the goal, informatics and genomics but not how these tools are used "in the field" and what are the trade offs of cost vs. the ability to accurately estimate parameters that underlie the work reported here.

·

Basic reporting

No comments

Experimental design

No comments

Validity of the findings

Major points
1. The most significant critique I have is that the application of the kSNP method to these data is insufficiently detailed and disparities with the other methods insufficiently discussed. Please clarify how pi and the number of segregating sites are estimated for the kSNP method. It isn’t clear if invariant kmers are included in aspects of the analysis but apparently they are or all sites would be segregating? Do overlapping kmers with different focal SNPs get accepted in the analysis, and does that influence the phase or allele frequency distribution of SNPs that get included? The most striking disparities I see in the data are associated with kSNPs: inferred substitution rates are substantially higher than WGS, but estimated pi is much lower. What biological explanation could there be for this? I would take these disparities as prima facie evidence of ascertainment biases associated with this application of the method, which could be explored by examining the locations of kSNP sites in the reference genomes and the allele frequency distributions for the two classes of genomic variants.
2. Lines 82-90. I find the language of hypothesis testing used here to be unnecessary and unconvincing. The first hypothesis amounts to ‘less data gives less resolution’, with the implicit alternative ‘less data gives equal resolution’. This is a straw man. The more useful question not asked here is ‘how many MLST loci are needed to approach the resolution of SNPs/WGS, and is that number cost-effective as a standard approach’? Or at the very least, does MLST as currently practiced produce erroneous conclusions (as distinct from incomplete ones)? The last sentence of this paragraph should be removed, the important point of evaluating relevant species having a range of expected genetic diversity and with different life histories has already been made. The authors cannot further claim the power to evaluate these as factors in a statistical sense. There is no replication that would allow for a general estimation of the impact of diversity/life history traits. Line 199: as before, ‘test’ does not seem accurate; this is an investigation of a few species likely to represent the genetic diversities of relevance to clinical molecular epidemiology, done in a largely qualitative manner. A useful comparison with real-world examples, but not a robust statistical test.
3. This is a comparison by several useful criteria, but it should be acknowledged that other criteria both for and against MLST use are not considered. For example, cost, infrastructure, and necessary expertise still favor MLST for some uses/users. On the other hand, WGS provides unique information about gene content not accessible to MLST and potentially not kSNPs as well. WGS, and MLST to a lesser extent, can provide haplotype information that kSNPs cannot, and WGS is well suited to statistics like dN/dS, Tajima’s D and Fu & Li’s D. Ultimately, this need not be an either/or question, and one could unequivocally favor genome-scale molecular epidemiological databases for all significant pathogens while noting also that WGS could be coupled with or informed by broader MLST surveys.
4. The major case of MLST discordance is for B. pseudomallei. The MLST substitution rate is, remarkably, substantially higher than for genomic snps, and accordingly, the inferred node ages are substantially shorter. There is little discussion as to why this pattern was so different from the other two species, and it is treated as two independent manifestations of discordance unique to MLST. This was also the smallest MLST data set in terms of both loci and bases.
5. Line 345: The suggestion that MLST is inherently flawed because of a violation of neutrality is dubious and my reading of the citation does not support it. It is not necessarily a problem for phylogenetic analysis to use selected loci to recover topology or estimate substitution rates (other than having less data), rather the potential problems relate to non-independence of sites (e.g., overestimated distance because multiple sites can be rapidly fixed) and parallelism among lineages. The vast majority of published molecular phylogenies are based on conserved genes under strong purifying selection most of the time with potential episodes of positive selection. It would of course be wise to remove genes evidencing divergent selection on the time scale of a given analysis, but it is reasonable to assume this is not the case for most MLST genes in use. It would also be reasonable to expect a large sample of SNPs to include sites under selection, so the “problem” is not unique to MLST.
6. The title “Evaluation Of Genomic Tools For Molecular Epidemiology” is not an accurate description of the work. Nothing so comprehensive is achieved, nor could be attempted in a brief communication. The language throughout the paper is of ‘molecular surveys’, so that is what should be used in the title. From my reading, the gist is that genome sequencing provides greater resolution of deeper nodes in molecular epidemiology and greater partitioning of variation by epidemiological factors, but there remains substantial concordance among the methods. Perhaps “Concordance and discordance of sequence survey methods for molecular epidemiology”?



Minor points
1. Lines 46-50 seem extraneous to the point of the paper, regardless of their validity. That there is an analysis bottleneck does not directly bear on concordance/discordance of different survey methods.
2. Line 219: Why should the number of segregating sites be equated to theta/effective population size in apparent distinction from nucleotide diversity? Both are estimators of theta under neutral equilibrium, although Watterson’s estimator is preferred. I would drop references to effective population size/theta and just state that segregating sites and pi are being estimated as measures of genetic diversity.
3. Line 211: Presenting the results for chr1 and chr2 separately is an unnecessary distraction that makes figures and tables busier without adding insight relevant to the paper, and is potentially confusing to readers unfamiliar with the unusual genome structure of those species. This is a general-audience journal.
4. Little guidance is provided as to how to interpret tree topology metrics. Do the numbers need to be scaled to node number of the trees? As a frame of reference, for the many almost equally likely trees usually recovered by ML for the same data set, what range of topology metrics might be observed?
5. Figure 1 seems supplemental in nature given the information in Table 1.
6. Table 2 is difficult to read at a glance. It would be helpful to just list the parameters/variables as row headers rather than using the table legend and formatting to distinguish them. Please be consistent in the use of significant digits, e.g. 4300 vs. 1931.65 is rather capricious use of decimal points.
7. It would be very helpful to include confidence intervals for segregating sites and pi, as there are formulas for the variance of these estimators and they are known to be inefficient estimators.

Additional comments

Although the authors are clearly experts in the field and the data are significant, the analysis is superficial at times and fails to substantially illuminate the questions of how much genetic variation is needed to achieve adequate resolution and how well different classes of loci represent lineage history. The authors also overreach as to the generality and robustness of their analysis, i.e. in the title and overuse of the language of hypothesis/statistical testing. The conclusions seem foregone, and what appear to me unexpected features of the data are unexplored. One could just as well argue that only seven loci under purifying selection producing two orders of magnitude less sequence (never mind the reduced cost, infrastructure, and expertise) produced reasonable results for B. pseudomallei, and it is little surprise that the WGS data sets perform better for the criteria that were examined. That kSNPs actually perform comparably to WGS in these contexts seems far from certain based on my reading.

---

## Round 0.2 · accepted · Accept

Thank you for your interesting submission and the revised version with comments, this will be an excellent paper for PeerJ